# Intestinal Barrier Permeability in Obese Individuals with or without Metabolic Syndrome: A Systematic Review

**DOI:** 10.3390/nu14173649

**Published:** 2022-09-03

**Authors:** Mariana Duarte Bona, Carlos Henrique de Medeiros Torres, Severina Carla Vieira Cunha Lima, Ana Heloneida de Araújo Morais, Aldo Ângelo Moreira Lima, Bruna Leal Lima Maciel

**Affiliations:** 1Institute of Biomedicine, Department of Medicine, Federal University of Ceara, Fortaleza 60430-270, Brazil; 2Postgraduate Medical Sciences Program, Department of Medicine, Federal University of Ceara, Fortaleza 60430-270, Brazil; 3Department of Nutrition, Center for Health Science, Federal University of Rio Grande do Norte, Natal 59078-970, Brazil; 4Postgraduate Nutrition Program, Center for Health Science, Federal University of Rio Grande do Norte, Natal 59078-970, Brazil; 5Postgraduate Biochemistry and Biology Molecular Program, Bioscience Center, Federal University of Rio Grande do Norte, Natal 59078-970, Brazil

**Keywords:** intestinal barrier permeability, markers, obesity, metabolic syndrome

## Abstract

Altered intestinal barrier permeability has been associated with obesity and its metabolic and inflammatory complications in animal models. The purpose of this systematic review is to assess the evidence regarding the association between obesity with or without Metabolic Syndrome (MetS) and alteration of the intestinal barrier permeability in humans. A systematic search of the studies published up until April 2022 in Latin America & Caribbean Health Sciences Literature (LILACS), PubMed, Scopus, Embase, and ScienceDirect databases was conducted. The methodological quality of the studies was assessed using the Newcastle–Ottawa scale (NOS) and the Agency for Healthcare Research and Quality (AHRQ) checklist. The Grading of Recommendations Assessment, Development and Evaluation (GRADE) framework was used to assess the quality of the evidence. Eight studies were included and classified as moderate to high quality. Alteration of intestinal barrier permeability was evaluated by zonulin, lactulose/mannitol, sucralose, sucrose, lactulose/L-rhamnose, and sucralose/erythritol. Impaired intestinal barrier permeability measured by serum and plasma zonulin concentration was positively associated with obesity with MetS. Nonetheless, the GRADE assessment indicated a very low to low level of evidence for the outcomes. Thus, clear evidence about the relationship between alteration of human intestinal barrier permeability, obesity, and MetS was not found.

## 1. Introduction

The obesity pandemic is a severe health problem because of related morbidity and costs [1]. Metabolic and cardiovascular complications are a major obesity-associated burden, especially for type-2 diabetes mellitus, metabolic syndrome, (MetS) and, in the long-term, cardiovascular diseases [2].

MetS includes the coexistence of physiological, biochemical, clinical, and metabolic factors associated with an increased risk of cardiovascular diseases [3]. It is characterized by the presence of three of five risk factors (high waist circumference, elevated plasma glucose, raised plasma triglycerides, reduced plasma high-density lipoprotein (HDL) cholesterol, and increased blood pressure), according to the National Cholesterol Education Program’s Adult Treatment Panel III (NCEP: ATP III) [4]. Studies have shown that obesity and MetS may be associated with changes in intestinal microbiota composition (dysbiosis), and this could induce impairment in intestinal barrier function, leading to a major impact on both immunological and metabolic functions in the host [5,6,7,8,9].

The integrity of intestinal barrier function is the result of ongoing equilibrium and crosstalk involving the gut microbiota, mucus, enterocytes, gut immune system, and gut-vascular barrier, allowing the permeability of essential ions, nutrients, and water but restricting the entry of toxins and potently pathogenic bacteria [10]. In experimental animal models of obesity, significant alterations in the intestinal barrier function occur, leading to increased intestinal permeability and favoring translocation of microbiota-derived lipopolysaccharide (LPS) to the bloodstream [11,12,13]. This results in a two- to threefold increase in LPS serum concentration in response to non-infectious stimuli, which is defined as metabolic endotoxemia [14]. Endotoxemia may trigger toll-like receptors (TLR) into 4-mediated inflammatory activation, eliciting a chronic low-grade proinflammatory and prooxidative stress status associated with obesity and MetS [15,16].

Clinical studies have observed a decrease in intestinal barrier permeability (IBP) after treatments for weight reduction in patients with obesity [17,18,19]. Another study has also demonstrated an increase in the IBP of individuals with type-2 diabetes mellitus, which was implicated in an important contribution to MetS [20].

The IBP can be measured by quantification of the passage of nondigestible markers which pass across the epithelial cell layer either between the epithelial cells (paracellular route) or through the cell membranes (transcellular route) [21,22]. These methods aim to analyze the flow of molecules from the intestinal lumen to the extraintestinal space, such as blood, specific organs, or urine, to evaluate intestinal barrier function integrity [23].

Sugar probes are the most common molecules used to test IBP and are ingested orally and dosed in urine after some time [24]. Among these probes, the most frequently used are lactulose and mannitol, which are passively absorbed from the gut without considerable metabolism and are unalterably excreted in the urine, directly correlating to their absorbed amount from the intestine [22]. Thus, an elevated lactulose/mannitol ratio suggests intestinal barrier dysfunction [25]. Other orally administered sugar probes used to evaluate the IBP are sucralose, sucrose, and rhamnose [26]. Sucralose is a sugar which is poorly absorbed in the human intestine and poorly degraded by the colonic microbiota as lactulose and mannitol; for this reason, it is useful in assessing colonic permeability [27]. Sucrose tests the gastroduodenal permeation [28], and rhamnose is used as a marker for small-bowel permeability [26]. In some studies, all these sugar probes are used simultaneously in order to appraise pan-gastrointestinal permeability [28].

As another IBP marker, zonulin is a protein involved in the modulation of intracellular tight junctions (TJ) and is implicated in the regulation of small intestine permeability by inducing the opening of TJ between epithelial cells [29]. In humans, high serum concentrations of zonulin were validated and correlated with increased IBP measured with ELISA [30].

These in vivo tests have immense potential in clinical research, as they provide a non-invasive real-life setting where the intestinal barrier function can be analyzed as a relevant factor in health and disease [31]. On the other hand, these tests are time-consuming, and several confounding factors can compromise the interpretation of results [32]. Individual differences in motility, intestinal cell surface, epithelial cell integrity, renal function, bacterial degradation, gastric dilution, and diet can influence in vivo intestinal permeability interpretations [33]. Moreover, variations in the methodologies, such as sugar solution concentration, urine collection period, assay method, and sensitivity can result in variations leading to missing standardized protocols [21]. Studies have suggested that assay duration, fasting periods, and consistency in the probes used could help reduce variations [34,35], reducing some in vivo confounding factors [36].

Although the association between impaired IBP and obesity and its metabolic disorders are well described for animal experimental models [5,6,7,8,9], studies in humans are still scarce. Evidence in humans of an association between altered IBP, obesity, and MetS could lead to treatments that correct the IBP, decreasing the cardiovascular risk factors associated with obesity and MetS [37,38].

Given the considerable clinical implications of altered IBP, obesity, and MetS, this study conducted a systematic review that aimed to investigate the presence of an association between obesity with or without MetS and altered IBP in humans. In addition, the systematic review evaluated assays and markers used in the studies regarding IBP methodologies.

## 2. Materials and Methods

### 2.1. Protocol Registration

The systematic review protocol was registered in the International Prospective Register of Systematic Reviews (PROSPERO) on 10 July 2020 (CRD42020178658) and is available at https://www.crd.york.ac.uk/prospero/display_record.php?ID=CRD42020178658. (accessed on 20 July 2022). The drafting of this manuscript, including the flow diagram used in the search strategy, adhered to the PRISMA (Preferred Reporting Items for Systematic Reviews and Meta-Analysis) guideline (Appendix A) [39]. The writing of the manuscript was based on the systematic review protocol published by Bona et al. [40], and the guiding question for the systematic search was: “Is there alteration of intestinal barrier permeability in individuals with obesity with or without metabolic syndrome?”.

### 2.2. Inclusion and Exclusion Criteria

This systematic review included observational studies published without language restriction in scientific journals that met the eligibility criteria [40] used to evaluate IBP in adults and/or elderlies with obesity with or without MetS diagnosis.

Review researches, systematic reviews, case reports, books, conference proceedings, short communications, editorials, clinical trials, randomized studies, letters to the editor, theses, dissertations, studies with animal models, studies with children or teenagers (which assessed IBP mainly in the context of infections and undernutrition), and studies that evaluated IBP in adults and/or elderlies with obesity and/or with other diseases such as hepatic or celiac diseases (because they could significantly impact the degree of inflammation) were excluded.

### 2.3. Search Strategy

A comprehensive virtual search of the literature from the last 15 years (2007–2022) was performed using the PubMed, LILACS, Scopus, EMBASE, and ScienceDirect databases. The search strategy is presented in Appendix A using keywords indexed in the Medical Subject Heading (MeSH) as a search strategy with high sensitivity. Articles were transmitted to the Mendeley Reference Manager (V.1.19.4), and duplicates were detected and deleted. Following the eligibility criteria, two authors (MB and CT) performed the initial screening of studies based on the information contained in their titles and abstracts, and subsequently a full-paper screening was conducted by the same independent investigators. Where the reviewers disagreed, a third reviewer made a final decision (BM). A screening of the references of the included articles was also performed to identify potentially eligible studies not found in the original database search.

### 2.4. Data Extraction

After choosing the studies to be included in the review, two reviewers elaborated independent Microsoft Excel spreadsheets with the data from these articles. The following information was extracted and summarized in the spreadsheet: study characteristics (title, authorship, year and language of publication, site where the study was conducted, and study design); population characteristics (health status, sample size, age, and gender of the participants); methods to evaluate intestinal permeability; description of the results; relevant conclusions; and reported limitations.

### 2.5. Methodological Quality

The methodological quality assessment and risk of bias was carried out by two independent trained reviewers (MB and CT) using the Newcastle–Ottawa scale (NOS) for case-control studies and the Agency for Healthcare Research and Quality (AHRQ) checklist for cross-sectional studies. A third reviewer resolved any disagreement (BM).

The NOS includes eight questions analyzing the studies in terms of selection of participants, comparability between the subjects, and verification of exposure. The questions are scored with “one” or “no stars,” and the sum of these stars classifies the article [41]. The AHRQ consists of eleven items with the options “Yes,” “No,” or “Unclear.” A score “0” is attributed to items evaluated with “No” or “Unclear,” and a score “1” for those evaluated with “Yes” [41]. To better present and unify the results, the scores evaluated by the NOS and the AHRQ were converted into three quality categories: 0–3 (low quality); 4–7 (moderate quality); and 8–11 (high quality), as proposed by Cabral et al. [42].

### 2.6. Best-Evidence Synthesis

The best-evidence synthesis was guaranteed and the risk of bias due to selective publication was controlled by a narrative synthesis, according to the review protocol [40] and assessment of the quality of evidence. The Grading of Recommendation, Assessment, Development, and Evaluation (GRADE) framework was used to assess the level of evidence regarding the association evaluated in this systematic review. GRADE ranks the evidence as high (when there is high certainty that the association is very unlikely to change); moderate (when there is moderate certainty that the association may not change); low (when there is limited certainty that the association may not change); and very low (when certainty in the association estimated is very limited, leading any finding to be uncertain) [43].

## 3. Results

### 3.1. Search Selection

The virtual search of the PubMed, LILACS, Scopus, EMBASE, and ScienceDirect databases, encompassing the full electronic search strategies (Appendix A), retrieved 21,752 records, along with four additional records identified through a manual search. After the exclusion of duplicates, 5876 records had their titles and abstracts screened, and 5802 were excluded due to not contemplating the eligibility criteria. Of the 75 articles selected for full-text assessment, 67 were excluded because they evaluated obesity with other metabolic disorders such as Nonalcoholic Fatty Liver Disease (NAFLD), celiac disease, and inflammatory bowel disease (*n* = 15); evaluated IBP due to the modulation of gut microbiota using prebiotics and probiotics (*n* = 9); evaluated IBP due to surgical intervention (*n* = 6); evaluated only intestinal microbiota but not IBP (*n* = 22); described the study design incorrectly (*n* = 10); or because the full text was not available despite an attempt being made to contact the corresponding author (*n* = 5). Therefore, 08 articles were included in the review. The flow diagram of the screening process is presented in Appendix A.

### 3.2. Studies and Population Characteristics

The characteristics of the included studies are presented in Table 1. The prevalent study design was the case-control (75%), and all the studies were published in English. Sample sizes varied from 24 [44] to 123 [45], and the majority were conducted in Europe, followed by South America. The age of participants ranged from 18 [44] to 75 [46], and female was the gender most prevalent.

### 3.3. Assessment of Intestinal Barrier Permeability

Regarding the evaluation of IBP in the studies, Table 2 presents the IBP markers used in each study, the results of the studies, and the scores (quality categories). Zonulin was the most frequently used marker, followed by lactulose/mannitol, sucralose, sucrose, lactulose/L-rhamnose, and sucralose/erythritol.

Lactulose/mannitol (L/M) was measured in urine samples after a collection period of 5 h using a gas chromatograph [44,47] and 6 h using urinary plasma chromatography/mass spectrometry [46]. In this last study, the participants fasted during the collecting period, but in the other studies the participants were allowed to eat after the solution ingestion. In all three studies, this marker showed no alterations between the groups. In only one study [47], the individual excretion of mannitol tended to be higher (*p* = 0.06). Individual excretion of lactulose was significantly higher in subjects with obesity (*p* = 0.04), and the lactulose/mannitol ratio was higher in subjects with obesity, but not statistically significant (0.0180 ± 0.008 vs. 0.0144 ± 0.006, *p* = 0.13). Also, in this study, subjects that showed L/M values above the median had lower HDL-cholesterol levels (*p* = 0.03), higher values of TC/HDL ratio (*p* = 0.02) and LDL/HDL ratio (*p* = 0.02), higher fasting insulin (*p* = 0.02), and a higher HOMA index (*p* = 0.01). This indicated an association between L/M ratio and variables related to metabolic risk factors.

Zonulin was measured in serum [45,50,51] and in plasma [49] using a competitive ELISA kit (Immundiagnostik AG, Bensheim, Germany), and the assay sensitivity was <0.01 ng/mL in these studies. In the study by Zak-Golab et al. [49], plasma zonulin level was correlated positively with body mass (R = 0.30, *p* < 0.01), BMI (R = 0.33, *p* < 0.01) and fat mass and percentage (R = 0.31, *p* < 0.01 and R = 0.23, *p* < 0.05, respectively). Also, the zonulin level was proportional to the daily energy intake (which was associated with obesity) and inversely proportional to the protein percentage dietary intake (which was associated with normal weight). Morkl et al. [51] observed a positive correlation between higher zonulin serum concentrations and BMI (R = 0.235, *p* = 0.017), total fat mass (%) measured with BIA (R = 0.205, *p* = 0.039), and waist and hip circumference (R = 0.263, *p* = 0.007 and R = 0.231, *p* = 0.202, respectively). In the study by Moreno-Navarrete et al. [45], circulating zonulin significantly increased in obese versus nonobese subjects (*p* = 0.007) and in subjects with glucose intolerance (*p* = 0.03). Circulating zonulin increased with BMI (*p* = 0.002), waist-to-hip ratio (WHR) (*p* = 0.025), fasting insulin (*p* < 0.001), fasting triglycerides (*p* = 0.02), and uric acid (*p* = 0.025), and was negatively associated with HDL-cholesterol (*p* = 0.02) and insulin sensitivity (*p* = 0.002). Finally, in the study by Mokkala et al. [50], a linear positive relationship was observed between higher zonulin concentration serum detected in women with obesity and elevated concentration of inflammatory markers (hs-CRP and GlycA), fasting insulin, HOMA2-IR, fasting triglycerides, and total and LDL-cholesterol; moreover, a negative correlation with insulin sensitivity was observed.

All other saccharides were measured in urine samples. In two studies [44,46], sucralose was administered and analyzed simultaneously with other saccharides. In one these studies [46], its recovery showed differences between the groups, which tended to increase in overweight subjects and increased significantly in obese as compared to nonobese subjects (*p* = 0.014). Sucrose was also administrated simultaneously with other saccharides and analyzed in urine after a collection period of 1 h [48] and after a collection period of 6 h [46]. Neither study showed any difference in sucrose recovery between the groups. Lactulose/L-rhamnose and sucralose/erythritol recovery also did not show any difference between the groups from the single study that evaluated these IBP markers [48].

### 3.4. Quality Assessment and Risk of Bias

Regarding the criteria for the classification of methodological quality, the studies included in this systematic review were classified as moderate to high quality, with the majority (62.5%) representing high methodological quality (Table 2).

### 3.5. Association between Intestinal Barrier Permeability and Obesity

The summary of the association between obesity and intestinal barrier permeability for each IBP marker is presented in Table 3. The summary of the evidence demonstrated very low quality based on the GRADE framework for almost all of the outcomes with the exception of zonulin.

Obesity was positively associated with alteration of IBP assessed by zonulin in four studies [45,49,50,51] with low quality of evidence due to reporting bias. Higher concentrations of zonulin were observed in subjects with obesity, which increased with body mass index (BMI), fat mass, and serum glucose in all the studies. Two studies [45,50] observed that subjects with an elevated zonulin concentration were positively associated with a higher concentration of many metabolic risk markers (triglycerides, fasting insulin, CRP, and LDL- and total cholesterol). Therefore, a positive association between alteration in IBP and obesity with MetS in the last two studies was observed, due to the correlation between zonulin and the raised metabolic and inflammatory markers concentration detected.

The alteration of the IBP assessed by the other markers was not associated with obesity. Three studies evaluated the alteration of the IBP using lactulose/mannitol (L/M) [44,46,47], and none observed any significant association or correlation with obesity. Nevertheless, in the study of Teixeira et al. [47], a higher L/M ratio was associated with elevated concentrations of metabolic markers in subjects with obesity. Thus, the L/M ratio could be associated with obesity with MetS.

IBP evaluated by sucralose was assessed in two studies [44,46], and only one [46] observed that abnormal colonic permeability was significantly associated with individuals with obesity. Sucrose, lactulose/L-rhamnose, and sucrose/erythritol excretion were analyzed, and there was no difference between individuals with or without obesity.

## 4. Discussion

Increased IBP was described in animal models of obesity in association with elevated endotoxemia and alterations in the glucose metabolism [11,12]. However, it remains unknown whether this also takes place in subjects with obesity, because studies measuring IBP in humans are scarce and their results inconsistent [52]. In the present systematic review, we investigated if obesity with or without MeS is associated with intestinal barrier function impairment. Moreover, the methodologies and markers measuring IBP were evidenced.

This review identified eight observational studies that matched the eligibility criteria. In three of these, the authors did not find alteration in IBP in obese subjects compared with nonobese subjects, and, in the others, there was an association between increased IBP markers and obesity. None of the studies evaluated obese subjects with MetS as a separate group. Nevertheless, three studies [45,47,50] reported increased IBP associated with elevated MetS variables such as HOMA index, waist circumference, and fasting triglycerides. Further studies evaluating the alteration of IBP in subjects with MetS as a separate group are necessary to establish if these previous results are consistent and if the IBP markers used are valid to evaluate this population.

Zonulin was analyzed in four studies [45,49,50,51], and a positive association between increased IBP, obesity, endotoxemia [49], immunoinflammatory markers [45,50,51], and lipid and glucose metabolism [45,50] was found. These studies used the same methodology with moderate-to-high quality assessment, differing only in the source of samples. However, the quality of the association evidence was low due to differences in the study design, to the results, to the low number of studies included, and to reporting bias. Moreno-Navarrete et al. [45] observed higher zonulin concentrations in the serum of individuals with impaired glucose tolerance. Therefore, they suggested that zonulin might help as a small intestine permeability marker for glucose intolerance and insulin resistance. These findings were not observed by Zak-Golab et al. [49], and this difference may be the result of the reported limitations of their study, namely, beyond sample size, the lower percentage of insulin-resistant individuals in the population of the study. Therefore, further studies are crucial to investigate the potential contribution of zonulin-associated loss of intestinal barrier function to obesity and associated glucose metabolism disturbance.

In the other four studies, the saccharides were detected simultaneously in urine samples collected for 5 h after solution ingestion, except for one study which took 6 h [46]. However, this collection time did not interfere with the results, and the methodologic quality assessment of the studies received a moderate [48] to high [46,47] rating. Lactulose/mannitol was analyzed in three studies [44,46,47]. A higher individual lactulose excretion in subjects with obesity was observed in only one study [47], indicating damage in the paracellular route, allowing a higher flux of molecules through the space between the enterocytes [22]. However, due to the low number of studies in the present systematic review and the inconsistency of the results, the summary of evidence is very low, and the IBP alteration measured by lactulose/mannitol needs additional studies with fewer limitations.

One study [46] observed that obesity was associated with colonic permeability measured by sucralose but without evidence of altered stomach and small-intestine permeability, and this was inconsistent with the results of the other two studies [44,48], which also analyzed sucralose excretion but did not observe any difference between the groups. Therefore, the quality of alteration of IBP measured by sucralose is also very low. Urinary recovery of the other saccharides was not associated with obesity or MetS (lactulose/L-rhamnose, sucrose, sucralose/erythritol).

Limitations of the present study include the fact that the review consisted of a heterogenous mix of studies and thus no meta-analysis was conducted. This is because the statistical procedure for association studies requires the reporting of different items controlled for multivariable analysis, and the original studies’ poor reporting lacked adequate data details [53]. Therefore, as a limitation, there was no assessment of quantitative data. The strengths of this systematic review are the methodological quality classification of the studies and the overall evaluation of the exposure evidence with each IBP marker measurement. To our knowledge, this systematic review is a pioneering study on this subject. Thus, based on the summary of evidence in this systematic review, there is not enough evidence to affirm that obesity with or without MetS in humans is associated with intestinal barrier impairment.

## 5. Conclusions

There is no definitive evidence on the association between obesity with or without MetS and increased IBP in humans, due to the small number of articles found and the low quality of the data. However, there was a positive association between alteration of IBP in individuals with obesity compared to those without obesity in studies using zonulin as an IBP marker. Moreover, an association between higher zonulin concentration and L/M ratio values with MetS variables was found in some studies. These results might indicate that subjects with obesity and MetS present impaired intestinal barrier function compared to subjects with obesity but without MetS. Therefore, further research with adequate design and minimum risk of bias must be performed for consistency of results and evidence.

## Figures and Tables

**Table 1 nutrients-14-03649-t001:** Characteristics of the studies included in this systematic review.

Study Design	Authors (Year)	Country	Sample Size	Gender Groups
Case-control	Brignardello et al. [44]	Chile	24	Male and female
	Moreno-Navarrete et al. [45]	Italy	123	Male
	Teixeira et al. [47]	Brazil	40	Female
	Verdam et al. [48]	The Netherlands	28	Male and female
	Zak-Golab et al. [49]	Poland	80	Male and female
	Di Palo et al. [46]	Italy	120	Male and female
Cross-sectional	Mokkala et al. [50]	Finland	100	Female
	Morkl et al. [51]	Austria	102	Female

**Table 2 nutrients-14-03649-t002:** Results of intestinal barrier permeability (IBP) evaluation and critical appraisal.

First Author (Year)	Studied Groups	IBP Markers	Samples	Results	Scores (Quality Categories)
Brignardello et al. [44]	11 lean and 13 obese	Lactulose/Mannitol Sucralose	Urine collected over a period of 5 h	There was no difference to both markers between the groups	8 (High)
Moreno-Navarrete et al. [45]	90 lean and 33 obese	Zonulin	Serum	Zonulin increased significantly in obese	6 (Moderate)
Teixeira et al. [47]	20 lean and 20 obese	Lactulose/Mannitol	Urine collected over a period of 5 h	Lactulose/mannitol was no significantly different between the groups, only lactulose individual excretion was significantly higher in the obese groups.	8 (High)
Verdam et al. [48]	13 lean and 15 obese	Sucrose Sucralose/Erythritol Lactulose/L-rhamnose	Urine collected over a period of 1 and 5 h	There was no difference between the groups	7 (Moderate)
Zak-Golab et al. [49]	30 lean and 50 obese	Zonulin	Plasma	Zonulin was significantly higher in obese	6 (Moderate)
Di Palo et al. [46]	45 lean, 30 overweight and 45 obese	Lactulose/Mannitol Sucralose Sucrose	Urine collected over a period of 6 h	Lactulose/mannitol and sucrose showed no difference between the groups. Sucralose increased significantly in obese	8 (High)
Mokkala et al. [50]	52 overweight and 48 obese	Zonulin	Serum	Higher serum zonulin were associated with subjects with obesity and MetS	8 (High)
Morkl et al. [51]	45 lean, 17 individuals with anorexia nervosa, 21 overweight and 19 obese	Zonulin	Serum	Higher serum zonulin were correlated with obeses with higher BMI	9 (High)

**Table 3 nutrients-14-03649-t003:** Summary of evidence of the associations between obesity with or without MetS and intestinal barrier permeability (IBP) evaluated for each IBP marker.

Exposure	Outcomes	Number of Studies (Participants)	Quality of Evidence	Evidence Summary
Obesity with or without MetS	Alteration of IBP—lactulose/mannitol	3 (184)	⊕●●● Very low due to inconsistency of results and bias	No association
	Alteration of IBP—lactulose/L-rhamnonse	1 (28)	⊕●●● Very low due to number of studies, sparse evidence and bias	No association
	Alteration of IBP—sucralose	2 (144)	⊕●●● Very low due to number of studies, inconsistency of result and bias	No association
	Alteration of IBP—sucrose	2 (148)	⊕●●● Very low due to number of studies, different design and bias	No association
	Alteration of IBP—zonulin	4 (405)	⊕⊕●● Low due to reporting bias	Positive association
	Alteration of IBP—sucralose/erythritol	1 (28)	⊕●●● Very low due to number of studies, and related limitations	No association

⊕●●●: very low grade; ⊕⊕●●: low grade; ⊕⊕⊕●: moderate grade; ⊕⊕⊕⊕: high grade.

## Data Availability

All the data from this study are available in the manuscript and in Appendix A.

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
