# Peer review of "Intestinal Barrier Permeability in Obese Individuals with or without Metabolic Syndrome: A Systematic Review"

_nutrients, 2022, doi:10.3390/nu14173649_

Round 1

Reviewer 1 Report

This manuscript provided a systematic review on the relationship between alteration of human intestinal barrier function and obesity, which is an interesting topic. This manuscript is well written. However, I have the following concerns:

1. There were only eight published papers included in this systematic review in consideration of many factors affecting the relationship between obesity and intestinal barrier function and the evaluation on the relationship as well as in considation of low quality of evidence, it is difficult to make a solid conclusion on the relationship, which makes this systematic review less significant.

2. Intestinal barrier permeability just indicates a part of intestinal barrier function, and this review is focused on intestinal barrier permeability, thus it is inappropriate to use intestinal barrier function in the title and major conclusion. Please revise it.

3. In Table 2, for the study of Mokkala et al, the authors just provided the information on the number of overweight individuals without numbers of lean and obese individuals in the field of studied groups, why? There is a difference between the definitions of overweight and obese This systematic review is focused on intestinal barrier function in obese individuals but not the overweight individuals.

4. Medication usually affects metabolism, especially for the metabolic disorders-related medication. Did the authors examined this for the studies included? If they did, I suggest that authors provide a statement about it in the manuscript.

5. This manuscript needs minor editing. For example, there are some typos, assessed was mistakenly typed as assed, plasma was mistakenly typed as plasme.

Author Response

We thank the reviewer for the kind analysis of our manuscript.

Reviewer 2 Report

I have now read and assessed the manuscript Nutrients-1863712 : Intestinal Barrier Function in Obese Individuals with or with out Metabolic Syndrome: A Systematic Review

-         -There are several typo, grammar and English errors that need to be corrected throughout the article.

-         - It was very difficult to concentrate on the review due to the bad English in which the manuscript has been written. I suggest the whole review has to be written by an English native speaker before re-submission to the journal.

-         - Cushing syndrome is one of the subset of MetS, but it is not mentioned. What is the situation of intestinal barrier permeability (IBP) in this case?

-          -Why all studies in these categories are excluded? “Studies with children 125 or teenagers, studies that evaluate intestinal barrier function in adults and/or elderly with 126 obesity with other diseases as hepatic or celiac diseases were excluded”

-        -  The number of studies that were considered in this systematic review is too small, and perhaps that is why a negative conclusion was obtained.

-        -  I do not understand the rationale of the Methodological quality the authors used. It seems to me to be confusing, inaccurate, and not scientific.

-          -Table 3 showed that 13 studies have been used. How that happened bearing in mind only 8 studies were included.

-          -I do not know why 67 full-text studies were excluded from the review bearing in mind among them some important diseases that influence the IBP.

-          I suggest the authors can re-assess their criteria of inclusion and exclusion and think of a better scoring method to determine the quality of work to be included. Then they can re-submit the review.

Author Response

(The authors gave the same response as above.)
